# Investigation of Galectin-3 and Cardiotrophin-1 Concentrations as Biomarkers in Dogs with Neurological Distemper

**DOI:** 10.3390/vetsci12050499

**Published:** 2025-05-20

**Authors:** Alper Erturk, Aliye Sagkan Ozturk, Atakan Ozturk

**Affiliations:** 1Department of Internal Medicine, Veterinary Faculty, Hatay Mustafa Kemal University, Antakya 31040, Hatay, Türkiye; 2Department of Molecular Biochemistry and Genetic (Medicine), Institute of Health Sciences, Hatay Mustafa Kemal University, Antakya 31040, Hatay, Türkiye; 3Department of Physiology, Medicine Faculty, Gaziantep Islam Science and Technology University, Gaziantep 27010, Gaziantep, Türkiye

**Keywords:** canine distemper virus, galectin-3, cardiotrophin-1, neurological distemper, biomarkers

## Abstract

Canine distemper is a highly morbid and fatal disease that manifests itself with both systemic and neurological signs. In recent years, biomarkers such as Galectin-3 and Cardiotrophin-1 have been studied in inflammatory and degenerative diseases. However, their role in neurological distemper has not been investigated yet. The study aimed to investigate the differences in serum concentrations of Galectin-3 and Cardiotrophin-1 between dogs with neurological distemper and a control group, and to explore their correlations with hematobiochemical parameters. Galectin-3 and Cardiotrophin-1 serum concentrations were found to be significantly elevated in dogs diagnosed with neurological distemper when compared to the control group. In addition, significant correlations were found between both Galectin-3 and Cardiotrophin-1 and certain hematological parameters. Galectin-3 and Cardiotrophin-1 may serve as potential biomarkers for the pathogenesis of neurological distemper.

## 1. Introduction

*Morbillivirus canis* (also known as canine distemper virus (CDV)), which belongs to the family *Paramyxoviridae* and genus *Morbillivirus*, is a cause of a multisystemic disease characterized by high morbidity and mortality worldwide [1,2,3]. It is most often seen in unvaccinated and undervaccinated dogs, some of which may present with neurological clinical signs due to viral-induced encephalopathy weeks to month after primary CDV infection [1,2]. Morbilliviruses are associated with severe systemic infection characterized by respiratory, gastrointestinal, and neurological signs. Approximately 50% of cases present with systemic signs without neurological involvement. Approximately 10 days after infection, CDV begins to spread from the primary sites of replication to various epithelial tissues and the central nervous system (CNS). Neurological manifestations of distemper may occur simultaneously with systemic signs or may develop 1 to 3 weeks after apparent recovery from systemic disease. Systemic signs include anorexia, fever, serous oculonasal discharge, cough, dyspnea, vomiting and diarrhea, which may occur in various combinations [4].

Viral activity in the CNS, particularly in young animals (<6 months), can cause acute encephalitis leading to gliosis, myelin degradation, demyelination, permanent neurological signs, and death [4,5,6]. Neurological signs include hyperesthesia, seizures, ataxia, paraparesis, tetraparesis, plegia, and myoclonus [4,7]. Old dog encephalitis is a rare subacute or chronic progressive panencephalitis thought to result from persistent CDV infection. Affected dogs are typically over 6 years of age and do not present with concurrent systemic signs. This type of encephalitis has not been seen in young dogs [4,5,6].

The canine distemper virus (CDV) is genetically similar to the human measles virus and rinderpest virus [3]. CDV is a major pathogen that has mutated from the measles virus, causing canine distemper and fatal meningitis. Although no cases of CDV infection in humans have been reported, its association with diseases such as Paget’s disease and multiple sclerosis (MS) has sparked discussions regarding its zoonotic potential [8,9]. Owing to the demyelination observed during neuropathological transformation, CDV is considered an important model for studying diseases such as MS in humans [5].

Galectin-3 (Gal-3) is a beta-galactoside-binding protein of the lectin family, involved in multiple biological processes including proliferation, differentiation, programmed cell death, intercellular adhesion, and tissue regeneration [10]. It plays a crucial role in inflammatory response processes involved in the pathogenesis of neuroinflammatory and neurodegenerative diseases [11,12]. In humans, Gal-3 concentrations are elevated in CNS damage such as Alzheimer’s disease, demyelination, and hypoxia/ischemia [12,13]. In addition, Gal-3 contributes to the neuropathological mechanisms underlying viral infections affecting the CNS [14,15]. Studies in veterinary medicine have investigated the role of Gal-3 in endocrine, dermatological, cardiac, and neoplastic diseases [16].

Cardiotrophin-1 (CT-1) belongs to the interleukin-6 family of cytokines. It is primarily generated in the myocardium and released into systemic circulation through the coronary sinus. It modulates inflammation and exerts protective effects on cardiac cells and the nervous system [17,18]. Its concentration may increase during inflammation in nerve cells [19,20]. It has been suggested that in humans, CT-1 may support the survival of damaged nerve cells, such as dopaminergic neurons in Parkinson’s disease [21]. Although there are no data on the use of CT-1 as a biomarker in veterinary medicine, data on its role in neurological diseases in humans are limited.

A better understanding of the pathogenesis of CDV is necessary for the development of effective prevention and treatment strategies. It has been hypothesized that CDV may induce neuroinflammation and demyelination, leading to increased concentrations of biomarkers Gal-3 and CT-1. The aim of the present study was to evaluate potential differences in Gal-3 and CT-1 biomarkers in blood serum between the neurological distemper and control group, and to investigate possible correlations between these biomarkers and hematobiochemical parameters in dogs with neurological distemper.

## 2. Material and Methods

The Institutional Ethics Committee of the Faculty of Veterinary Medicine, Hatay Mustafa Kemal University, granted ethical approval for this study (Ref. No. 2025/02-01).

### 2.1. Study Groups

This study was conducted on 19 owned dogs presented to the Hatay Mustafa Kemal University Veterinary Health Application and Research Center. Thirteen of these dogs were in the neurological distemper group, aged between 2 and 6 months, unvaccinated, and of different breeds and sexes (eight males, five females; five mixed breeds, three Pointers, three Rottweilers, one German Shepherd, and one Anatolian Shepherd). Six dogs (four males, two females; four crossbreeds, one Pointer, and one Boxer) of the same age group brought for vaccination and routine examinations were included in the control group based on clinical examination and complete blood count (CBC) analyses.

### 2.2. Clinical Examination

All dogs included in the study underwent a comprehensive systemic examination, and neurological signs (behavioral changes, seizures, cerebellar signs, vestibular signs, visual loss, paresis, paralysis, and myoclonus) were recorded. Fecal analysis was performed on all dogs included in the study. Dogs with a history of exposure to toxins or drugs (ketamine, alfaxalone, morphine, and fluralaner, as well as metabolic toxins associated with conditions like hepatic encephalopathy, uremic encephalopathy, kernicterus, and mycotoxins) that could potentially cause myoclonus were excluded from this study.

### 2.3. Rapid Diagnostic Test Applications

Conjunctival swabs were collected from dogs with a preliminary diagnosis of neurological distemper, based on clinical examination findings and a rapid diagnostic test (Asan Easy Test CDV Ag^®^, ASAN Pharm. Co., Ltd., Hwaseong, Republic of Korea; relative sensitivity, 97.96%; relative specificity, 97.50%). Dogs with positive test results and clinical findings were assigned to the neurological distemper group. The same analyses were performed for the control group, and those who were negative were included in the study.

### 2.4. Collection of Blood Samples

Blood samples were collected from the dogs aseptically through the antebrachium cephalic vein. K_3_-EDTA tubes were used for CBC analysis, whereas anticoagulant-free tubes were used for serum biochemistry and biomarker analysis. Serum samples were allowed to clot at room temperature for 15 min and then centrifuged for 10 min at 4000× *g*. Serum was harvested and stored at −80 °C until analysis.

### 2.5. Complete Blood Count Analysis

White blood cell (WBC) count, lymphocyte percentage (Lymph %), granulocyte percentage (Gran %), monocyte count, platelet count (PLT), mean platelet volume (MPV), platelet distribution width (PDW), and plateletcrit (PCT) were measured using a complete blood count analyzer (BC-2800 Vet, Mindray, China).

### 2.6. Serum Biochemical Analyses

Creatinine, blood urea nitrogen (BUN), total protein, albumin, total bilirubin, alanine aminotransferase (ALT), lactate dehydrogenase (LDH), alkaline phosphatase (ALP), and creatine kinase (CK) levels were measured via an automated biochemistry analyzer (Chem 200vet, Gesan, Italy).

### 2.7. Measurement of Serum Galectin-3 and Cardiotrophin-1 Concentrations

Blood serum Gal-3 (Bioassay Technology Laboratory (BT Lab)^®^, Cat. No. E0343Ca, Jiaxing, China) and CT-1 (Bioassay Technology Laboratory (BT Lab)^®^, Cat. No. E0461Ca) concentrations were measured using commercial canine-specific ELISA kits according to the manufacturer’s instructions. The measurable sensitivity of Gal-3 is 0.26 ng/mL, and the test range for Gal-3 concentrations is between 0.5 ng/mL and 200 ng/mL. The measurable sensitivity of CT-1 was 2.39 ng/L, and the test range for CT-1 concentration was between 5 ng/L and 1000 ng/L.

### 2.8. Statistical Analysis

IBM SPSS Statistics 25 software was utilized for the data analysis. The Shapiro–Wilk test was used to determine whether the data followed a normal distribution. An independent two-sample test was performed for the normally distributed data. The Mann–Whitney U test was used for the data that did not follow a normal distribution. Differences between the groups were considered statistically significant at *p* < 0.05. Correlations between Gal-3 and CT-1 concentrations and other parameters were evaluated using Spearman’s rho correlation, with asterisks indicating significance (* *p* < 0.05, ** *p* < 0.01).

## 3. Results

### 3.1. Clinical Examination Findings

Clinical signs, such as paralysis, paraparesis, spastic tetraparesis, tetraplegia, facial and masticatory muscle twitching (chewing spasms), muscle atrophy, and myoclonus, were observed in the dogs in the neurological distemper group. Non-epileptic and recurrent generalized or localized myoclonus (four dogs with myoclonus in the head only, four dogs with myoclonus in the pelvic limbs only, two dogs with myoclonus in the head and pelvic limbs, and three dogs with myoclonus in the head and all four limbs) was observed in all dogs. Anorexia, cachexia, fever, depression, and conjunctivitis were observed in 11 dogs, while no abnormalities other than neurological signs were observed in two dogs. Extraneural signs such as respiratory signs, fever, and conjunctivitis were observed in eight dogs approximately 20 days prior to development of neurological deficits, while no systemic signs were observed in five dogs. The onset of neurological signs was gradual and progressive in all dogs. No parasites were found during fecal examination of the dogs included in the study.

### 3.2. Hematobiochemical Results

The lymphocyte and monocyte percentages were lower in dogs from the neurological distemper group, whereas AST enzyme activity and Gran % were higher than those in the control group (*p* < 0.05) (Table 1).

### 3.3. Biomarker Analysis

The Gal-3 and CT-1 concentrations were significantly higher in the neurological distemper group than in the control group (*p* < 0.05) (Table 2). The correlation of these parameters with the hematobiochemical parameters is presented in the heatmap and scatterplot in Figure 1. A negative correlation was found between Gal-3 and monocytes (*p* < 0.05) and a positive correlation was found between Gal-3 and PLT and PCT (*p* < 0.05). A negative correlation was found between CT-1 and Lymph % (*p* < 0.01). However, a positive correlation was found between CT-1 and Gran % (*p* < 0.01).

## 4. Discussion

This study investigated the relationship between CNS damage and inflammation by evaluating the serum concentrations of biomarkers such as Gal-3 and CT-1 in dogs with neurological distemper. This study suggests that Gal-3 and CT-1 concentrations were significantly higher in dogs with neurological distemper and that these biomarkers correlated with some hematobiochemical parameters. This suggests that Gal-3 and CT-1 may serve as biomarkers for the understanding of neurological distemper pathogenesis.

After CDV viremia, around day 10 of the infection, the virus spreads to the epithelial tissues in CNS via hematogenous routes or cerebrospinal fluid [5]. Neurological signs may develop during or after the systemic phase of the disease, even in the absence of systemic signs [5]. Myoclonus is a typical sign of encephalomyelitis in young dogs with affected neurological systems [22]. The clinical findings observed in the present study were consistent with those reported in the literature [5,22,23].

Although various hematobiochemical parameters have been reported in CDV infections, they are not sufficient for the specific diagnosis of neurological distemper [4,24,25]. The lymphopenia, monocytopenia, and granulocytosis observed in neurological distemper in this study may be associated with viral replication, immunosuppression, and inflammation [4,5,25].

In virus-infected conditions, Gal-3 plays a complex role in host–pathogen interactions, influencing both immune responses and tissue damage. During viral infection, it regulates viral entry and spread and is highly expressed in immune cells, fibroblasts, epithelial cells, and endothelial cells [26,27]. Gal-3’s role in viral pathogenesis is both complex and diverse [28]. It can contribute to tissue damage and inflammation in certain viral infections, but it may also provide protective effects. This highlights its potential as both a therapeutic target and a biomarker [27]. Studies in Gal-3 knockout mice have shown reduced lung damage and inflammation during influenza A virus infection [27]. Furthermore, Gal-3 is involved in the pathogenesis of neurodegenerative and neuroinflammatory diseases such as MS, Alzheimer’s, Parkinson’s, and Huntington’s disease. Gal-3 levels correlate with the severity of these diseases [11]. Neuropathological lesions, such as traumatic and ischemic brain injury, demyelination, encephalitis, and demyelination, cause microglial activation and produce proinflammatory Gal-3 [12,29,30].

Gal-3 may disrupt the blood–brain barrier and trigger the production of cytokines involved in the pathogenesis of neurodegenerative diseases [31]. In mice exposed to hypoxia/ischemia at birth, elevated Gal-3 concentrations were observed in the damaged areas of the brain. Furthermore, Gal-3 deficiency results in reduced tissue loss in the hippocampus and striatum [32]. Jiang et al. [33] showed that deletion of Gal-3 in mice attenuated the severity of autoimmune encephalomyelitis, a widely used animal model of myelin degeneration in the CNS. However, Gal-3 expression is increased in microglia that have phagocytose myelin compared to microglia that do not [11]. In a previous study, Gal-3 concentrations were elevated in dogs with meningoencephalitis of unknown etiology compared to healthy dogs [34]. In contrast, Gal-3 plays a critical role in the recovery of inflammatory demyelinating disorders by maintaining myelin integrity and function by promoting oligodendrocyte differentiation [35]. As Gal-3 can exert both pro and anti-inflammatory effects in the CNS, the inhibitors developed suggest Gal-3 as a therapeutic target in CNS diseases [11]. Considering and understanding the role of Gal-3 in viral infections may facilitate the development of novel treatment and prevention strategies.

In the present study, a negative correlation (*p* < 0.05) between Gal-3 and monocytes and a positive correlation (*p* < 0.05) between PLT and PCT were detected. Gal-3 may exacerbate the proinflammatory state by enhancing monocyte/macrophage chemotaxis, promoting monocyte migration, and macrophage infiltration into tissues [36]. Chen et al. [36] identified Gal-3 as a potential candidate that plays a role in thrombogenesis by acting as a novel positive regulator of platelet hyperreactivity and thrombus formation in the individuals diagnosed with coronary artery disease. Gal-3 is thought to cause this condition by binding to platelet Dectin-1 and activating the Dectin-1/Syk signaling pathway. In the same study, TD139, a Gal-3 inhibitor, improved atherothrombosis and myocardial infarction by specifically inhibiting platelet hyper-reactivity. In the present study, the observed correlations between Gal-3 and various hematological parameters provide further evidence supporting its role in promoting proinflammatory responses and thrombogenesis. These findings suggest that Gal-3 may exert both proinflammatory and anti-inflammatory effects within the central nervous system, potentially influencing the balance between tissue damage and protective mechanisms under neuroinflammatory conditions.

Cardiotrophin-1 is expressed in the brain, thymus, lungs, kidneys, liver, intestines, testes, prostate, skeletal muscle, adipose tissue, and the cardiovascular system [18]. In fact, CT-1 has been reported to exert a synergistic effect with glial-derived neurotrophic factor in the development of nervous tissue, protection of the mature nervous system against various injuries and dysfunctions, and the survival of motor neurons [19,37]. In particular, the protective effect of CT-1 on dopaminergic neurons, achieved by inhibiting tyrosine hydroxylase activity and promoting choline acetyltransferase activity, is not mediated by other members of the IL-6 family [38]. Although most members of the IL-6 family present a hydrophobic N-terminal secretion signal sequence, CT-1 lacks this secretion sequence. This is similar to the glial-derived neurotrophic factor. This suggests that CT-1 acts as a neuroregulatory cytokine that prevents neuronal damage in the nervous system [19,38,39]. In another study, a 50–75% decrease in the number of astrocytes was found in the neonatal cortex of CT-1 deficient mice [40]. Adenovirus-mediated administration of recombinant Cardiotrophin-1 (rCT-1) prevents motoneuron cell death and long-term motor axon degeneration in a mouse model of progressive motor neuropathy. Furthermore, CT-1 inhibited muscle denervation and improved neuromuscular function in this mouse model [20]. CT-1 also provided therapeutic benefits in terms of functional and morphological parameters in a mouse model of spinal muscle atrophy [41,42]. Adenovirus-mediated CT-1 gene transfer and rCT-1 administration delayed neurogenic muscle atrophy and progressive neuromuscular disorders in an experimental model of amyotrophic lateral sclerosis [41,43]. In an experimental epilepsy model, the transplantation of neural stem cells modified with CT-1 significantly diminished the frequency of recurrent seizures, likely by facilitating neuronal repair and regeneration [44]. Transgenic mouse models of Alzheimer’s have shown a marked reduction in CT-1 expression in the hippocampus, and restoration of CT-1 tissue concentrations in these animals has been shown to lead to a remarkable improvement in cognitive function [45].

In the present study, the positive correlation between CT-1 and Gran % can be attributed to the fact that CT-1, a cytokine belonging to the interleukin-6 family, plays a crucial role in modulating the inflammatory response by increasing both the number and activity of granulocytes. This elevation in granulocyte levels is typically observed as part of the body’s response to inflammation. Conversely, the negative correlation between CT-1 and Lymph % may reflect the suppressive effect of elevated CT-1 production during inflammatory processes. This impairs the functionality of lymphocytes involved in immune responses. Specifically, CT-1 affects the Th1 response in CD4+ T lymphocytes, which plays a key role in regulating immune cell activity [46].

In light of these findings, the increased concentration of CT-1 in dogs with neurological distemper may be linked to potential improvements in neuromuscular function and neuroprotective properties of this cytokine. Furthermore, as highlighted by Adams et al. [47] collaboration between veterinarians and medical professionals remains essential to unravel the underlying mechanisms of demyelinating diseases in both animals and humans. This study suggests a valuable foundation for future research to explore the role of CT-1 as a potential therapeutic target for the treatment of neurological disorders.

This study had several limitations. The sample size was relatively small. Secondly, there was no serial monitoring of changes in Gal-3 and CT-1, so only a single measurement was made. In addition, the association of biomarker concentrations with clinical factors such as survival rates and prognosis could not be compared. Another limitation is the lack of corrective assays using other markers of neuronal damage, such as neuron-specific enolase.

## 5. Conclusions

In conclusion, this study suggested that Gal-3 and cardiotrophin-1 concentrations may be elevated in dogs with neurological distemper, suggesting their potential as biomarkers in veterinary neurology. These biomarkers may be useful in assessing neuroinflammation and monitoring disease progression. These findings may contribute to a better understanding of disease mechanisms and may inform future efforts in biomarker development and clinical application.

## Figures and Tables

**Figure 1 vetsci-12-00499-f001:**
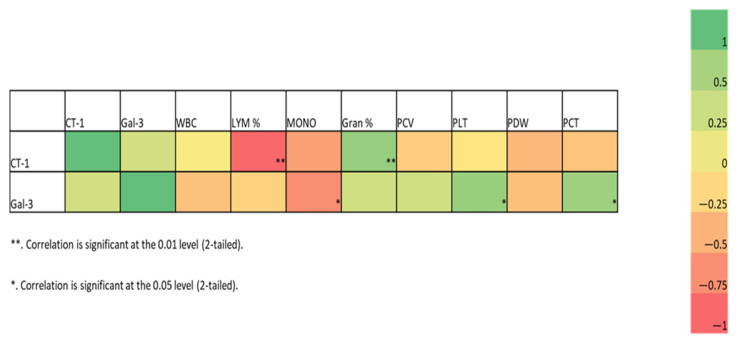
Correlation heatmap and scatterplot. The heatmap shows the Spearman correlation coefficients between all parameters for the combined C/D groups. The colors represent the correlation (−1 to +1), with red indicating more negative and green more positive correlations. * (*p* < 0.05), ** (*p* < 0.01). CT-1: Cardiotrophin-1; Gal-3: Galectin-3; WBC: white blood cell; LYM%: lymphocyte percentage; MONO: monocyte count; Gran%: granulocyte percentage; PCV: packed cell volume; PLT: platelet count; PDW: platelet distribution width; PCT: platelecrit.

**Table 1 vetsci-12-00499-t001:** Hematobiochemical parameters of control and neurological distemper dogs.

	Control Group (Mean ± SD)	Neurological Distemper (Mean ± SD)	*p*-Value
WBC m/mm^3^ (5.0–19.0) *	13.91 ± 2.52	12.32 ± 16.59	0.820
Lymphocyte % (5.0–30.0) *	29.82 ± 15.82	10.41 ± 3.0	0.003 *
Granulocyte % (40.0–80.0) *	60.64 ± 12.30	86.94 ± 3.29	0.000 *
Monocyte cells/mL (0.1–1.1) *	0.72 ± 0.25	0.32 ± 0.42	0.047 *
PLT m/mm^3^ (211–621) *	334.00 ± 171.76	536.44 ± 313.99	0.176
MPV fl	8.75 ± 0.33	8.68 ± 1.11	0.881
PDW	15.22 ± 3.08	15.20 ± 2.73	0.991
PCT %	0.29 ± 0.15	0.40 ± 0.21	0.298
Creatine mg/dL (0.3–1.4) *	0.64 ± 0.25	0.59 ± 0.42	0.776
BUN mg/dL (14–36) *	16.33 ± 5.24	15.23 ± 10.72	0.816
Total protein g/dL (4.0–5.8) *	4.93 ± 0.8	5.38 ± 1.28	0.438
Albumin g/dL (2.5–3.9) *	3.14 ± 0.4	2.99 ± 0.5	0.594
Total bilirubin mg/dL (0.1–1.0) *	0.6 ± 0.2	0.4 ± 0.2	0.285
ALT U/L (10–100) *	24.67 ± 9.93	21.85 ± 7.47	0.499
LDH U/L (20–500) *	160.83 ± 189.85	247.92 ± 214.67	0.407
ALP U/L (75–450) *	136.83 ± 83.60	125.46 ± 100.30	0.813
CK U/L (50–450) *	247.58 ± 206.36	308.39 ± 152.42	0.479

*, reference values; WBC: white blood cell; PLT: platelet, MPV: mean platelet volume; PDW: platelet distribution width; PCT: platelecrit; BUN: blood urea nitrogen; ALT: alanine aminotransferase; AST: aspartate aminotransferase; LDH: lactate dehydrogenase; ALP: alkaline phosphatase; CK: creatine kinase. Note: * *p* < 0.05.

**Table 2 vetsci-12-00499-t002:** Gal-3 and CT-1 concentrations in the control and neurological distemper of dogs.

	Control Group (Mean ± SD)	Neurological Distemper (Mean ± SD)	*p*-Value
Galectin-3 (ng/mL)	9.56 ± 1.32	12.61 ± 1.90	0.009 *
Cardiotrophin-1 (ng/L)	29.96 ± 13.41	44.64 ± 10.46	0.024 *

Note: * *p* < 0.05.

## Data Availability

The datasets used and/or analyzed during the current study are available from the corresponding author upon reasonable request.

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
