# Peer review of "Investigation of Galectin-3 and Cardiotrophin-1 Concentrations as Biomarkers in Dogs with Neurological Distemper"

_vetsci, 2025, doi:10.3390/vetsci12050499_

Round 1

Reviewer 1 Report

Comments and Suggestions for Authors

The manuscript authored by Erturk and colleagues presents findings on the investigation of Galectin-3 and Cardiotrophin-1 concentrations as biomarkers in dogs with neurological distemper.

Therefore, this manuscript by Erturk et al. has the major findings:

i: The serum levels of Galectin-3 and Cardiotrophin-1 were significantly higher in dogs with neurological distemper compared to the control group;

ii: Significant correlations were found between Galectin-3 and Cardiotrophin-1 and certain hematological parameters;

iii: Galectin-3 and Cardiotrophin-1 may serve as potential biomarkers for the pathogenesis of neurological distemper.

The authors' study is significant in the context of analyzing potential biomarkers for the severity of canine distemper. Additionally, the analysis of these biomarkers could aid in disease prognosis. However, as noted below, I have several suggestions that could further enhance the study.

Major Questions

  1. First of all, my question is whether it was not possible to track the outcome of the dogs with neurological distemper? It would be interesting to know if the levels of Galectin-3 and Cardiotrophin-1 were associated with a worse disease prognosis.

  1. Another interesting point would be to determine whether elevated levels of Galectin-3 and Cardiotrophin-1 are persistent in the sick dogs (with or without CDV positivity)... Did the authors have time-staggered samples to conduct this type of analysis?

Minor Questions

  1. Abstract (line 23): The new classification of the viral species by the ICTV is Morbillivirus canis.

  1. Introduction (line 44): Please considerer writing “Morbillivirus canis (also known as canine distemper virus (CDV))” instead of “Canine distemper virus (CDV)”

  1. Introduction (line 42): Please considerer writing “Paramyxoviridae” and genus “Morbillivirus” in italic.

  1. Material and Methods (lines 99-100): Clarify this sentence by providing examples of toxins and drugs that could introduce confounding biases in the analysis variable.

  1. Material and Methods (lines 125-130): There is limited information in the literature regarding the kits used. Therefore, it would be helpful to provide a more detailed description of the methodology. For example, were the samples measured in triplicates? How was the cutoff point determined?

  1. Discussion: It is important for the authors to conclude with a paragraph that addresses the limitations of the study. For examples: the reduced sample size between the groups evaluated (i.e., 13 vs 6), correction analysis with other markers of neuronal damage (i.e., Neuron-specific enolase...).
Comments on the Quality of English Language

English is acceptable, but will need to be revised after corrections.

Author Response

Please review the attached file

Reviewer 2 Report

Comments and Suggestions for Authors

The study entitled “Investigation of Galectin-3 and Cardiotrophin-1 Concentrations as Biomarkers in Dogs with Neurological Distemper” addresses a clinically significant and underexplored topic—biomarker evaluation in canine neurological distemper. The focus on Galectin-3 and Cardiotrophin-1 is novel within the veterinary context. The authors clearly state their objectives and provide a rational background, citing relevant literature on neuroinflammation and potential biomarker roles in both human and veterinary settings. Use of ELISA for biomarker quantification is appropriate and validated. The control and disease groups are reasonably well defined. Ethical approval and methodological transparency are adequately addressed. The statistical methods (Mann-Whitney U, Spearman correlation) are suitable for the small sample size and non-normally distributed data. The discussion provides a solid integration of current findings with the broader literature, including relevant insights into neuroinflammation and immune modulation.

However, these aspects should be improved

Areas for Improvement

  1. Several sentences are awkwardly constructed, repetitive, or contain grammatical errors. For example: "Gal-3 and CT-1 may be potential biomarkers of the pathogenesis of neurological distemper." Could be revised to: "Gal-3 and CT-1 may serve as biomarkers for the diagnosis and understanding of neurological distemper pathogenesis." Phrases and ideas are sometimes repeated across abstract, results, and discussion (e.g., describing correlations multiple times). Aim for concise synthesis. Terms such as "neurological distemper" and “dogs with distemper” are used interchangeably. Standardize to avoid confusion.
  2. Figure 1 (Correlation Heatmap and Scatterplots): The heatmap is informative and useful for visualizing correlation patterns. Axis labels and parameter names are too small and abbreviated. Use full or clearer abbreviations. Color scale: Consider including a legend for correlation strength (e.g., -1 to +1). Scatterplots would benefit from regression lines or clearer significance notations.
  3. Tables 1 and 2: Formatting: Currently cluttered; align column headers for better readability. Statistical Annotation: Add asterisks or superscripts next to significant values in the tables (e.g., p<0.05).
  4. Sample Size Justification: The number of animals (13 distemper, 6 controls) may be low. While common in veterinary studies, a discussion of statistical power and potential limitations in generalizability would strengthen the manuscript.
  5. Reference Formatting: Some DOIs are inconsistently formatted (e.g., some missing spaces or improperly linked). Make sure all references follow the journal's style guide precisely.
  6. Conclusion Section: The conclusion is somewhat repetitive of the results and discussion. Consider: Highlighting novel contributions explicitly. Emphasizing clinical relevance and directions for translational application or future biomarker development.
  7. Minor Suggestions: Clarify the manufacturer details of the ELISA kits (e.g., provide full company name/location once). Avoid phrases like “our study believes” — rephrase to “this study suggests.” Add graphical abstract or visual summary to improve accessibility and attract broader readership.

Overall Recommendation

Major revision — the study has clear scientific merit and novelty but requires moderate revision for clarity, language polish, figure refinement, and structural improvements to enhance scientific rigor and reader comprehension.

Comments on the Quality of English Language
  1. Several sentences are awkwardly constructed, repetitive, or contain grammatical errors. For example: "Gal-3 and CT-1 may be potential biomarkers of the pathogenesis of neurological distemper." Could be revised to: "Gal-3 and CT-1 may serve as biomarkers for the diagnosis and understanding of neurological distemper pathogenesis." Phrases and ideas are sometimes repeated across abstract, results, and discussion (e.g., describing correlations multiple times). Aim for concise synthesis. Terms such as "neurological distemper" and “dogs with distemper” are used interchangeably. Standardize to avoid confusion.
  2. Major revision — the study has clear scientific merit and novelty but requires moderate revision for clarity, language polish, figure refinement, and structural improvements to enhance scientific rigor and reader comprehension.

Author Response

Please review the attached file

Reviewer 3 Report

Comments and Suggestions for Authors

See attached file.

Author Response

Please review the attached file

Round 2

Reviewer 1 Report

Comments and Suggestions for Authors

The authors have made modifications based on my suggestions. Therefore, I agree with this version of the manuscript.